# Mesencephalic Locomotor Region and Presynaptic Inhibition during Anticipatory Postural Adjustments in People with Parkinson’s Disease

**DOI:** 10.3390/brainsci14020178

**Published:** 2024-02-15

**Authors:** Carla Silva-Batista, Jumes Lira, Daniel Boari Coelho, Andrea Cristina de Lima-Pardini, Mariana Penteado Nucci, Eugenia Casella Tavares Mattos, Fernando Henrique Magalhaes, Egberto Reis Barbosa, Luis Augusto Teixeira, Edson Amaro Junior, Carlos Ugrinowitsch, Fay B. Horak

**Affiliations:** 1Exercise Neuroscience Research Group, University of São Paulo, São Paulo 05508-070, Brazil; 2Department of Neurology, Oregon Health and Science University, Portland, OR 97239, USA; horakf@ohsu.edu; 3School of Arts, Sciences and Humanities, University of São Paulo, São Paulo 03828-000, Brazil; fhmagalhaes@usp.br; 4School of Physical Education and Sport, University of São Paulo, São Paulo 05508-030, Brazil; 5Biomedical Engineering, Federal University of ABC, São Bernardo do Campo 09210-170, Brazil; danielboari@gmail.com; 6Centre for Neuroscience Studies, Queen’s University, Kingston, ON K7L 3N6, Canada; 7Department of Radiology, University of São Paulo, São Paulo 05508-090, Brazil; nuccimar@gmail.com (M.P.N.);; 8Instituto de Ciencias e Tecnologia, Universidade Federal de São Paulo, São Paulo 09913-030, Brazil; 9Movement Disorders Clinic, Department of Neurology, School of Medicine, University of São Paulo, São Paulo 05508-070, Brazil

**Keywords:** mesencephalic locomotor region, anticipatory postural adjustment, presynaptic inhibition, freezers, step initiation, H-reflex

## Abstract

Individuals with Parkinson’s disease (PD) and freezing of gait (FOG) have a loss of presynaptic inhibition (PSI) during anticipatory postural adjustments (APAs) for step initiation. The mesencephalic locomotor region (MLR) has connections to the reticulospinal tract that mediates inhibitory interneurons responsible for modulating PSI and APAs. Here, we hypothesized that MLR activity during step initiation would explain the loss of PSI during APAs for step initiation in FOG (freezers). Freezers (n = 34) were assessed in the ON-medication state. We assessed the beta of blood oxygenation level-dependent signal change of areas known to initiate and pace gait (e.g., MLR) during a functional magnetic resonance imaging protocol of an APA task. In addition, we assessed the PSI of the soleus muscle during APA for step initiation, and clinical (e.g., disease duration) and behavioral (e.g., FOG severity and APA amplitude for step initiation) variables. A linear multiple regression model showed that MLR activity (R^2^ = 0.32, *p* = 0.0006) and APA amplitude (R^2^ = 0.13, *p* = 0.0097) explained together 45% of the loss of PSI during step initiation in freezers. Decreased MLR activity during a simulated APA task is related to a higher loss of PSI during APA for step initiation. Deficits in central and spinal inhibitions during APA may be related to FOG pathophysiology.

## 1. Introduction

Over half of the individuals with Parkinson’s disease (PD) develop freezing of gait (FOG) [1]. FOG is one of the most debilitating features of PD that causes falls and poor quality of life [2]. Individuals with PD and FOG (freezers) present an inability to step forward despite the intention to walk [3]. The transition between the quiet stance and step initiation requires anticipatory postural adjustments (APAs) [4]. APAs are abnormal in freezers and are associated with FOG severity [5]. Freezers have delayed step initiation associated with repetitive APA [6], as if they were unable to inhibit their postural preparation and release the stepping program [3]. It has been hypothesized that freezers may have the inability to inhibit their postural state and initiate stepping [5,6]. This deficit in step initiation has been proposed to result from a lack of central inhibition [7].

We investigated the involvement of the spinal cord during postural preparation to initiate a step in young and older adults [8]. Presynaptic inhibition (PSI) is a spinal inhibitory mechanism often proposed to explain changes in the reflex pathways [9]. PSI mechanism involves GABAergic primary afferent depolarization interneurons in the spinal cord [9,10]. PSI is responsible for modulating sensory feedback at the spinal level for walking [11] and postural preparation [5,8,12,13,14]. Increases in the PSI levels may decrease Ia afferent inputs onto motoneurons during posture and gait, through activation of GABAergic primary afferent depolarization interneurons [14] that are controlled from supraspinal motor tracts [15]. In animal models, the precision of skilled movements (reach trajectory and velocity of the forepaw to the food) depends on sensory feedback and its refinement by GABAergic interneurons, as a higher presynaptic control is required during the precision of skilled movement [10].

Previous studies demonstrated higher PSI levels of soleus muscles during standing on a foam mat, which is consistent with a high proprioceptive feedback demand in healthy individuals [16]. We hypothesized that older compared to young adults require a higher presynaptic control of the soleus muscle to compensate for impaired supraspinal modulation on poor APAs. We found higher levels of PSI of the soleus muscle during impaired APA for older compared to young adults [8]. Higher PSI inhibition levels were associated with decreased APA amplitude. Like older adults, freezers can be thought to compensate for the lack of central inhibition during APA for step initiation using a higher presynaptic control.

We recently demonstrated that during impaired APAs, freezers have a loss of PSI of the soleus muscle compared to non-freezers and age-matched healthy controls, whereas the other groups have PSI during APAs [5]. The loss of PSI of the soleus muscle during step initiation was correlated with impaired APAs (i.e., small amplitudes) and FOG severity in freezers, suggesting that the lack of central inhibition of locomotor regions would be reflected in the loss of PSI of the soleus muscle during impaired APAs for step initiation in freezers.

Freezers compared to non-freezers have a dysfunction in the locomotor network that involves the mesencephalic locomotor region (MLR), the supplementary motor area (SMA), the subthalamic nucleus, and the cerebellar locomotor region [17]. Freezers compared to non-freezers showed higher functional connectivity between SMA and MLR and between the SMA and cerebellar locomotor region, which indicate that the brain cannot compensate for the lack of automatic control of gait by the basal ganglia [17]. The abnormal functional connectivity between MLR and SMA was associated with FOG severity [17].

MLR [18] and SMA [19,20] have neurons involved in the APA regulation. MLR [21] and SMA [22] send projections to reticulospinal neurons, which are known to regulate APA in an animal model (cat) [23]. Reticulospinal neurons also mediate PSI during fictive locomotion in an animal model (cat) [15]. Thus, the loss of PSI of the soleus muscle during step initiation found only in freezers would be associated with abnormal MLR and SMA activity during step initiation.

Therefore, this study aimed to identify which locomotor brain region (MLR, SMA, subthalamic nucleus, and cerebellar locomotor region) during a functional magnetic resonance image (fMRI) protocol of the APA task for step initiation would explain the loss of PSI of the soleus muscle during the APA task for step initiation. We used the APA task in an event-related fMRI protocol validated by our group [24,25]. We included these locomotor brain regions as they presented the beta change of the blood oxygenation level-dependent (BOLD) signal during the fMRI protocol of the APA task for step initiation. The beta change is a proxy of change in brain activity during the task, as we previously published [17,24,25].

Freezers have decreased BOLD signal within the MLR but not within the SMA during an fMRI protocol that simulates walking, which has been correlated with FOG severity [26]. Thus, we hypothesized that MLR activity during an fMRI protocol of the APA task for step initiation would explain the loss of PSI of the soleus muscle during APA for step initiation in freezers.

## 2. Materials and Methods

### 2.1. Ethical Approval

The University of Sao Paulo (USP) Ethical Committee (School of Physical Education and Sport—ref. 2011/12) approved the study, which was also registered at the National Clinical Trial (RBR-83VB6B). The study was performed in agreement with the Declaration of Helsinki. All individuals provided written informed consent.

### 2.2. Participants

Freezers diagnosed according to the UK Brain Bank criteria [27] were recruited from the Movement Disorders Clinic in the School of Medicine at the USP. Inclusion criteria were as follows: (1) stable dopaminergic therapy for at least two months before and during the experimental period; (2) presenting FOG during ON-medication state (scored >1 on the New FOG Questionnaire [28] and identified by a movement disorders specialist by videos of objective tests, such as turning clockwise and counter-clockwise); (3) Hoehn and Yahr stages 3–4; (4) 49–85 years of age; (5) able to walk safely without walking aids; (6) no physical exercise practice in the three months preceding study commencement; (7) Mini-Mental State Examination >23 [29]; (8) absence of other neurological disorders, significant arthritis, musculoskeletal or vestibular disorders; (9) absence of severe tremor, claustrophobia, and metal in the body; and (10) high quality of brain volumes acquired during the fMRI (head motion < 1 mm) [30].

### 2.3. Study Procedures

For this study, we used clinical, behavioral, PSI of the soleus muscle for step initiation, and fMRI data from our previous study only for freezers, as only they presented loss of PSI of the soleus muscle during step initiation and performed the fMRI protocol for step initiation [5,25]. In all assessments (clinical, behavioral, PSI, and fMRI), freezers were assessed in the ON-medication state within 1.5 to 2 h of taking their morning dose of dopaminergic medication.

### 2.4. Outcome Assessments

#### 2.4.1. Clinical Assessments

Clinical assessments included motor severity measured using the Unified PD Rating Scale motor subsection (UPDRS-III) [31], disease duration (years since diagnosis), levodopa-equivalent daily dosage scores calculated according to standardized methods [32], subjective FOG assessed by the New FOG Questionnaire scores [28], and cognitive inhibition assessed using the Stroop Color-Word Test—Victoria version [33].

#### 2.4.2. Behavioral Assessments

FOG severity using the FOG-ratio during a 2-min turning task, as previously published [34].

APA amplitude and duration for step initiation, as previously published [5,8]. Briefly, the onset of the APA (the time between the abrupt increase of the mediolateral force amplitude and the onset of the step) and the APA duration (the time between the onset of APA and the onset of the step) during step initiation were measured with the force platform (AMTI ORG-7). Mediolateral force amplitude during the step task was normalized by the distance between the malleoli of the individual (N/cm).

Electromyography (EMG) and co-contraction ratio, as previously published [5,8]. Briefly, self-adhesive surface disc EMG electrodes (1 cm in diameter) placed on the soleus and tibialis anterior muscles were used to record the EMG signals. The reference electrode was placed on the skin over the patella. The EMG signals were amplified (×1000) and bandpass filtered (10–1000 Hz) and stored on the computer of the Nicolet Viking Quest portable EMG apparatus (CareFusion, San Diego, CA, USA). We analyzed the rectified and averaged EMG recordings during the APA task that were measured over a 100 ms epoch that preceded tibial nerve stimulation (test H-reflex) or the common peroneal nerve stimulation (conditioned H-reflex), as previously described [5,8]. A co-contraction ratio was also calculated to express the rectified and averaged EMG amplitude for tibial anterior muscle relative to the rectified and averaged EMG for soleus muscle.

#### 2.4.3. Test and Conditioned H-Reflexes

We induced the soleus H-reflex by stimulating the posterior tibial nerve in the left leg via a monopolar stimulating electrode (1 ms rectangular pulse) over the popliteal fossa using a constant-current stimulator (Nicolet Viking Quest portable EMG apparatus, CareFusion, San Diego, CA, USA). The anode was placed proximally to the patella. Two self-adhesive surface disc surface EMG electrodes (1 cm in diameter) placed on the soleus muscle were used to record H-reflexes, with interelectrode distance of 3–4 cm. The peak-to-peak amplitude of the soleus H-reflex was used to measure the reflex responses. Intervals of 10 s were used to evoke H-reflexes. Stimulus intensities were increased in steps of 0.05 mA, starting below the soleus H-reflex threshold and increasing up to supramaximal intensity to measure the M_max_. The sensitivity of the soleus H-reflex to inhibitory and facilitatory effects depends critically on its size [35]. Then, we evoked the soleus H-reflex at an intensity corresponding to 20–25% of maximal motor response, which resulted in a soleus H-reflex on the ascending portion of its recruitment curve [36].

We evoked PSI of the soleus H-reflex by conditioning stimulation (1 ms rectangular pulses) of the common peroneal nerve through bipolar surface electrodes (0.5 cm in diameter). These electrodes were placed 1–3 cm distal to the neck of the fibula in the left leg [5,8,11,37,38]. Motor responses were recorded using two self-adhesive surface disc electrodes (1 cm in diameter) placed on the tibialis anterior muscle. We used an interval of 100 ms between the conditioned H-reflex and the H-reflex test. Our previous works had shown that the H-reflex is strongly inhibited using a conditioning-test interval of 100 ms [5,8]. Previous works showed that conditioning-test intervals of <100 ms are likely to involve postsynaptic mechanisms, decreasing the ability to assess presynaptic influences [39]. Also, recommendations have been proposed for studies that use soleus H-reflex depression by common peroneal nerve stimulation at a motor threshold level, which is indicated for conditioning-test intervals of 60–120 ms [9,40]. At a conditioning-test interval of 100 ms, stimulation of the common peroneal nerve evokes an inhibition that is attributed to PSI [11,37,41,42]. A stimulation intensity of 1.1 × motor threshold is submaximal for activation of all inhibitory interneurons [12,35,37,43]. Different interneurons transmit PSI to Ia terminals projecting to several motoneuron pools [9]. We checked that the stimulation evoked a motor response in the tibialis anterior muscle without a motor response in the peroneal muscles. Also, we ensured that the conditioning stimulus was applied at a position where the threshold for a motor response (motor threshold) in the tibialis muscle was lower than the motor threshold in the peroneal muscle. Additionally, we reduced the bias in the amount of inhibition (PSI), maintaining constant the size of the test H-reflex during the data collection. The soleus H-reflex is often depressed in the quiet stance [11,13,44], so, the test stimulus intensity during the APA was adjusted so that the reference (unconditioned) H-reflex attained the same size as in the quiet stance (our control condition) [5,8].

#### 2.4.4. Assessment of PSI of the Soleus Muscle during APA for Step Initiation

The force platform (AMTI ORG-7) was used to detect the abrupt increase of mediolateral force amplitude during APAs. Thus, when the APA amplitude exceeded 10–20% of the mean of the mediolateral force (corresponding to 2 standard deviations above the mean of the baseline force) an electrical stimulus (test or conditioned H-reflex) was automatically triggered. We used the LabVIEW software (5.1), as previously published [5,8], to calculate the baseline force threshold.

#### 2.4.5. Beta of the BOLD Signal Change of Locomotor Regions during Step Initiation

A detailed description of the APA task in the event-related fMRI protocol is included in the Appendix A. The beta of the BOLD signal change of locomotor regions of interest (SMA, subthalamic nucleus, MLR, cerebellar locomotor region), known to initiate and pace gait, were measured during an event-related fMRI protocol of step initiation These locomotor brain regions presented the beta change of the BOLD signal during the fMRI protocol of the APA task for step initiation [25]. The beta change is a proxy of change in brain activity during the task [24,25]. We evoked the conditioned and H-reflex test in the left leg in all participants (all participants initiated stepping with the right leg), the same leg required during the initiation of APA inside the scanner. Freezers tend to show predominant involvement of right-sided brain circuitry [7,45,46,47], which reinforces the importance of the APA task lateralized to the left leg. In addition, the participants had either both sides affected (moderate to severe PD—stages 3 and 4) or the left side affected (Table 1).

The 3.0 T MR system (Achieva, Philips Medical Imaging, The Netherlands, Amsterdam) with 32-channel head coil (80 mT/m gradient maximum amplitude) was used to obtain the images BOLD-sensitive images were acquired using T2*-weighted gradient echo-planar imaging (EPI): TR, 2.000 ms; TE, 30 ms; 40 slices; 3.0-mm slice thickness; 0.3-mm interslice gap; 3.0-mm^3^ isotropic voxels; 240 volumes (acquisition time, 8 min). Anatomical T1-weighted 3-D images were used for reference and image registration (T1-FFE; TR, 7 s; TE, 3.2 s; 180 slices; Flip angle, 8°; 1 mm^3^ isotropic voxels). fMRI data were processed using FSL software version 6.0 (www.fmrib.ox.ac.uk/fsl/, accessed on 30 April 2019) [48]. The volumes were preprocessed with an algorithm designed to reduce head movement (MCFLIRT), spatial smoothing (FWHM, 5 mm), [49] and images were affine registration to the standard space of MNI-152 (12 DoF) [50,51]. The level of head motion was less than 1 mm to avoid erroneous inference on neuronal function [30]. MCFLIRT is a motion correction tool based on FLIRT (FMRIB Linear Image Registration Tool), which is a fully automated robust, and accurate tool for linear (affine) intra- and inter-modal brain image registration used in FEAT (FMRI Expert Analysis Tool) from FSL. Further, the calculated motion parameters by the MCFLIRT tool during the pre-procedure were added in the statistic model as confounders/covariates, using the standard option, which considered only 6 parameters of motion correction (3 for translation and 3 for rotation) during the analysis, in addition to the task-related regressors [50,52]. The event of interest was the time from the onset of the first stimulus until 1 s after leg lifting onset, as detailed previously [24] and also included in the Appendix A. A linear model was implemented to estimate the BOLD signal in the event of interest compared to resting periods. The beta of the BOLD signal change of regions of interest was extracted using the featquery tool processing routine from FSL. The peak coordinates of the regions of interest of the right hemisphere during step initiation were SMA: x = 3, y = −13, z = 61, radius = 8 mm [26]; subthalamic nucleus: x = −11, y = −14, z = −3, radius = 8 mm [17]; MLR: x = 6, y = −30, z = −19, radius = 6 mm [17]; cerebellar locomotor region: x = 7, y = −52, z = −16, radius = 6 mm [17], which are illustrated in Figure 1.

### 2.5. Statistical Analyses

The normality and homogeneity of the data were tested by the Shapiro–Wilk and Levene’s tests, respectively. We used logarithm transformation for the beta of the BOLD signal change and PSI of the soleus muscle data, and we achieved normality. Then, one-tailed Pearson correlation coefficients were calculated among the variables as follow: clinical (UPDRS-III score, disease duration, medication dosage, clinical FOG score, and cognitive inhibition), behavioral (objective FOG severity, APA amplitude, APA duration, rectified and averaged EMG amplitude for tibial anterior and soleus muscle, and co-contraction ratio), PSI of the soleus muscle during step initiation, and the beta of the BOLD signal change of locomotor regions during step initiation (MLR, SMA, subthalamic nucleus, and cerebellar locomotor region). Then, we performed a linear multiple regression (PROC REG of SAS) using the stepwise method for PSI of the soleus muscle during step initiation as the dependent variable. To explain the variance of the dependent variable in the regression model, we used the clinical, behavioral, and beta of the BOLD signal change of regions of interest as independent variables. We also included age and height as independent variables, as these variables are known to be related to the H-reflex [53,54]. To avoid collinearity, we included the independent variables in the linear multiple regression analysis if they presented a *p*-value ≤ 0.05 and a correlation lower than 0.7 among them [55]. We used the SAS 9.2^®^ (Institute Inc., Cary, NC, USA) to perform the statistical procedures at *p* ≤ 0.05 (significance level). Results were presented as mean and standard deviation (SD).

## 3. Results

### 3.1. Participants

Forty freezers performed baseline testing (clinical, behavioral, PSI of the soleus muscle for step initiation, and an fMRI protocol for step initiation), with six of them being excluded from the analysis as detailed in Figure 2. The demographic, clinical, and behavioral variables, as well the values of PSI of the soleus muscle for step initiation and the beta of the BOLD signal change of regions of interest are presented in Table 1.

### 3.2. MLR Activity and APA Amplitude during Step Initiation Explain the Loss of PSI of the Soleus Muscle for Step Initiation

Freezers presented loss of PSI during APA for step initiation indicated by a high ratio of the conditioned H-reflex relative to the test H-reflex = 115.1 (8.6).

MLR activity (beta of BOLD signal change) during step initiation (r = −0.56, *p* = 0.0003), APA amplitude (r = −0.44, *p* = 0.0042), FOG-ratio (r = 0.50, *p* = 0.0011), disease duration (r = 0.49, *p* = 0.0016), and cognitive inhibition assessed using the Stroop Color-Word Test Stroop (r = 0.31, *p* = 0.0332) were associated with the loss of PSI of the soleus muscle during step initiation. These variables (except the Stroop Color-Word Test Stroop) entered the linear multiple regression model; however, only the MLR activity during step initiation (R^2^ = 0.32, *p* = 0.0006) and decreased APA amplitude (R^2^ = 0.13, *p* = 0.0097) significantly explained together 45% of the loss of PSI of the soleus muscle during step initiation in freezers, as demonstrated in Table 2. Figure 3 illustrates the association of loss of PSI with MLR activity (panel 3A) and APA (panel 3B).

## 4. Discussion

This is the first study to show the relationship among MLR activity, APA amplitudes, and loss of PSI of the soleus muscle, all assessed during step initiation in freezers. Although clinical (disease duration) and behavioral (FOG severity during turning) variables were entered in the regression model to explain the loss of PSI of the soleus muscle during step initiation, only MLR activity and APA amplitudes during step initiation explained together 45% of the loss of PSI of the soleus muscle during step initiation in freezers.

### 4.1. Why Do MLR Activity and APA Amplitudes Explain the Loss of PSI of the Soleus Muscle during Step Initiation in Freezers?

MLR activity and APA amplitude during step initiation may play an important role in the relationships between loss of PSI of the soleus muscle during step initiation and FOG of PD. The lack of central inhibition would be reflected in decreased MLR activity, as MLR receives a strong inhibition from basal ganglia due to the loss of dopaminergic neurons [56]. MLR is an important locomotor center of the midbrain [57]. Abnormal MLR inhibition would impair supraspinal motor tracts (e.g., reticulospinal tract) mediating inhibitory interneurons modulating PSI during APA in freezers. Freezers have decreased BOLD signal within the MLR during an fMRI protocol that simulates walking, which has been correlated with the FOG severity [26]. Freezers also have grey matter atrophy in the MLR [58]. MLR when stimulated increases postural tone for standing and induces stepping and running in a decerebrate cat [59,60]. MLR lesions cause cataplexy, akinesia in rats, and immobility attacks reminiscent of the FOG events in PD [57]. MLR [18], as well as the reticulospinal tract [23], has neurons involved in APA regulation. MLR sends projections to reticulospinal neurons [21] mediating PSI during fictive locomotion in an animal model (cat) [15]. Reticulospinal neurons receive short-latency orthodromic input from the MLR [21]. Thus, decreased or absent inputs of MLR may not be exciting the reticulospinal tracts, which are very important for regulating postural muscle tone and locomotion [61]. The reticulospinal tract modulates the activity of interneurons and motoneurons in spinal segments during posture and locomotion [62,63,64]. Spinal interneurons with GABAergic axo-axonic synapses on primary afferent terminals produce PSI, which regulates the sensorimotor drive during skilled movements in mouse [10]. In the decerebrate cat, medullary reticular formation induced generalized motor inhibition [65] and was associated with PSI of primary afferents [66]. These inhibitory effects were mediated by inhibitory interneurons [66,67,68]. Spinal GABAergic interneurons mediating PSI are modulated by the reticulospinal tract during locomotion in cats, independent of sensory feedback [15]. This finding suggests that the reticulospinal tract may program a different movement pattern modulating presynaptic control to adjust APA during stepping.

The pontomedullary reticular formation in the brainstem plays an important role in the control of posture and locomotion [69]. The pontomedullary reticular formation is the main source of the reticulospinal tract [69]. The reticulospinal tract is crucial for movement control with an important hub for sensorimotor integration, thus allowing cortical and subcortical structures to appropriately couple voluntary actions with posture and locomotion, like in APA [70]. APAs are worse in freezers than non-freezers and healthy controls [5], and APA has been found to be impaired in freezers [5,17,71]. Our previous study demonstrated that decreased APA amplitude during step initiation is associated with FOG severity and loss of PSI during step initiation [5]. Here, APA amplitude explained 13% of the loss of PSI, which suggests that impaired APAs may be due to abnormal reticulospinal tract projections on spinal interneurons that modulate PSI during APA for stepping in freezers [70].

As illustrated in our hypothetical model in Figure 4, glutamatergic projections from MLR are known to activate both inhibitory and excitatory pathways of the reticulospinal tract from pontomedullary reticular formation during postural control, gait, and locomotion in cat [61,72]. These glutamatergic projections may not be activating both inhibitory and excitatory pathways of the reticulospinal tracts due to MLR atrophy in freezers [7,58] or decreased MLR activity during walking [26]. Inhibitory reticulospinal tract is known to inhibit interneurons and motoneurons via inhibitory interneurons in the spinal cord in cats [61,65,73]. Thus, reticulospinal tract neurons may not be inhibiting GABAergic interneurons that mediate PSI of soleus Ia afferent terminals. As a result, the conditioned H-reflex is facilitated/increased (i.e., not inhibited), leading to loss of PSI during step initiation in freezers that are associated with impaired APA amplitude and FOG severity. We hypothesize that loss of central inhibition (abnormal MLR activity) may be reflected in the loss of spinal inhibition (PSI) for stepping in freezers.

Interestingly, the SMA, a cortical region that contributes to generating self-initiated and multi-segmental voluntary movements [74], did not enter into the regression model to explain the loss of PSI during step initiation in freezers. SMA sends motor commands to the reticulospinal tract [22], which coordinates APAs with step initiation [24]. The reticulospinal tract projects to the spinal motoneuron pool [72,75], sending drives to excitatory and inhibitory interneurons, mediating PSI during voluntary movements [15,26,41,76,77,78]. SMA has cortico-reticular projections to MLR [61,72,79]. Unlike MLR, SMA is hyperactive in freezers [80]. Freezers have increased SMA activity during APAs compared to non-freezers [24]. Freezers have increased functional connectivity between SMA and MLR compared to non-freezers [17]. The increased functional connectivity between SMA and MLR [17] would increase the excitability of the cortico-reticular projection arising from SMA on MLR activity. However, the increased connectivity of SMA on the MLR would increase the glutamatergic projections from MLR that are known to activate both inhibitory and excitatory pathways of the reticulospinal tract from the pontomedullary reticular formation during the posture, gait, and locomotion in cat [79]. MLR receives strong projections from the basal ganglia [56]. Overactivity of the output nuclei of the basal ganglia may lead to excessive paroxysmal inhibition of the already impaired MLR in freezers [56]. It is possible that inhibition of the basal ganglia on MLR is stronger than the increased excitability of SMA on MLR. Inhibition on MLR via basal ganglia has a negative and stronger influence on PSI, before triggering FOG, instead of hyperactivity of SMA on MLR. This is an issue open to future investigation.

It is important to highlight that older people and individuals with PD are often over-cautious and stay motionless due to fear of falling [81,82], which could impact PSI levels during step initiation. Increased postural stiffness and agonist–antagonist contraction is required to maintain postural stability during fearful situations [83]. PSI is known to increase during agonist–antagonist contraction and increase muscle stiffness during an upright stance in challenging situations (e.g., eyes closed) [16]. Since freezers have loss of PSI during step initiation and we did not find any association between co-contraction and loss of PSI, we believe that behavioral issues did not influence the PSI levels during step initiation in freezers. Thus, loss of PSI during step initiation may be due to abnormal MLR activity.

### 4.2. Future Directions for Treatment Strategies

We described some therapies that would improve MLR activity and PSI levels in freezers of PD, such as pedunculopontine nucleus-deep brain stimulation (an invasive therapy), spinal cord stimulation (a semi-invasive therapy), gene therapy, and rehabilitation therapies.

Our findings show that MLR may explain the loss of PSI during APA in freezers. MLR has been implicated in FOG pathophysiology [56]. Cellular loss within the MLR is associated with disease progression [84,85], which may explain why gait dynamic stability is affected by PD and is not responsive to levodopa [86]. We hypothesize that impaired MLR neurons likely lead to less activity of last-order interneuron via the reticulospinal tract [15], and consequently the loss of PSI in freezers. Pedunculopontine nucleus-deep brain stimulation seems to be an effective treatment option for severe PD, although its results in the literature are not conclusive [87]. Pedunculopontine nucleus-deep brain stimulation seems to mediate effects on the descending reticulospinal control, as bilateral pedunculopontine nucleus-deep brain stimulation alone or plus subthalamic nucleus improved spinal cord excitability (soleus H-reflex) of six individuals with advanced PD up to normal values of healthy controls [88]. Future studies should investigate the effects of pedunculopontine nucleus-deep brain stimulation on PSI levels in freezers.

Epidural spinal cord stimulation, a semi-invasive method, was investigated as a treatment option for gait disorders in PD [89,90]. Although spinal cord stimulation improved gait, APA duration [91], and FOG episodes [91,92], evidence is still inconclusive as these findings were recorded in a small number of individuals [89,90]. It has been suggested that this therapy may activate multiple structures along the somatosensory pathway responsible for the manifestation of PD symptoms [93,94]. In addition to stimulating specific somatosensory pathways, this therapy may also recruit brainstem arousal systems [93,94], such as MLR, that send motor commands to the spinal cord to initiate locomotion, via reticulospinal pathways [95]. This therapy may restore PSI levels in freezers. Future studies should test this hypothesis.

Loss of PSI in freezers suggests that GABAergic interneurons are not activated to inhibit the H-reflex. GABAergic interneurons form axo-axonic contacts with the central terminals of sensory afferents, exerting PSI over sensorimotor transmission [10]. Increased PSI [12] is a result of decreased Ia afferent inputs onto motoneurons through activation of GABA-ergic primary afferent depolarization interneurons [14]. GABA is abundant in the spinal cord, it is presented by interneurons only [96]. GABA-ergic interneurons are localized largely in the superficial laminae, while some are in deeper laminae of the dorsal and ventral horn [97]. Subthalamic AAV-GAD (adeno-associated glutamic acid decarboxylase) injection improved motor signs in hemiparkinsonian macaques [98] and in individuals with PD with Hoehn and Yahr stage 3 or greater [99]. The inhibitory neurotransmitter GABA is synthesized by GAD, which is predominant in the ventral horn of the spinal cord [100]. GAD is the key enzyme involved in the synthesis of the inhibitory neurotransmitter GABA from excitatory glutamate [101]. The AAV-GAD approach would be important in the spinal cord to restore PSI and improve FOG. Delivery of the gene encoding GAD could increase local GABA production within the spinal cord, restoring the function of GABA-ergic interneurons that mediate PSI.

We developed the exercise rehabilitation called resistance training with instability for PD, in which freezers PD [25] exercise with load/weight on unstable devices (a BOSU ball placed on the bases of support of individual), increasing sensorimotor integration. In non-freezers, we observed that 12 weeks of resistance training with instability were effective in increasing PSI levels at rest up to normal values of healthy controls [38]. In freezers, we observed increased MLR activity and improved APA amplitude after the 12 weeks of resistance training with instability [25]. Our next step is to verify whether this intervention can restore the PSI levels in freezers, as resistance training with instability is a sensorimotor intervention supposedly activating descending pathways that modulate the PSI [25].

Finally, recent studies have demonstrated the benefit of vibration (100–120 Hz) on FOG, suggesting this strategy as a novel therapy for freezers [102,103,104]. Vibration is an external somatosensory cue that involves enhanced proprioceptive processing while the vibration is provided in the feet or wrists [102,103,104]. The signals resulting from vibration ascend the spinal cord, which may reach the cortico–subcortical brain areas (e.g., thalamus and sensorimotor cortices) and interact with the motor system to improve gait [105]. Interestingly, PSI is responsive to vibration in healthy individuals [106,107,108]. Thus, this therapy can be assumed to have a potential to restore PSI levels and improve APA and MLR activity in freezers.

### 4.3. Limitations

This study has some limitations. First, all individuals were assessed in the ON medication state. Although ON medication assessments do not reflect the true disease state and FOG is most commonly seen in the OFF-state [109], ON medication decreases bradykinetic effects on FOG episodes [110]. Second, although fMRI imposes a restrictive environment, with limitations to assessing usual step initiation in a standing position, fMRI is the gold standard for in vivo imaging of the human brain to assess cortical and subcortical areas and presents high spatial resolution and optimal signal-to-noise ratio [111]. We used our fMRI protocol that simulates step initiation [24,25], which was validated to APA outside and inside the scanner [24]. Third, our participants had no FOG episode, thus we do not know whether the loss of PSI and the MLR activity would change during FOG episodes.

## 5. Conclusions

Decreased MLR activity during a simulated APA task is related to a higher loss of PSI during APA for step initiation. MLR activity and APA amplitudes during step initiation explained together 45% of the loss of presynaptic inhibition during step initiation in freezers. Deficits in central and spinal inhibitions during APA may be related to FOG pathophysiology.

## Figures and Tables

**Figure 1 brainsci-14-00178-f001:**
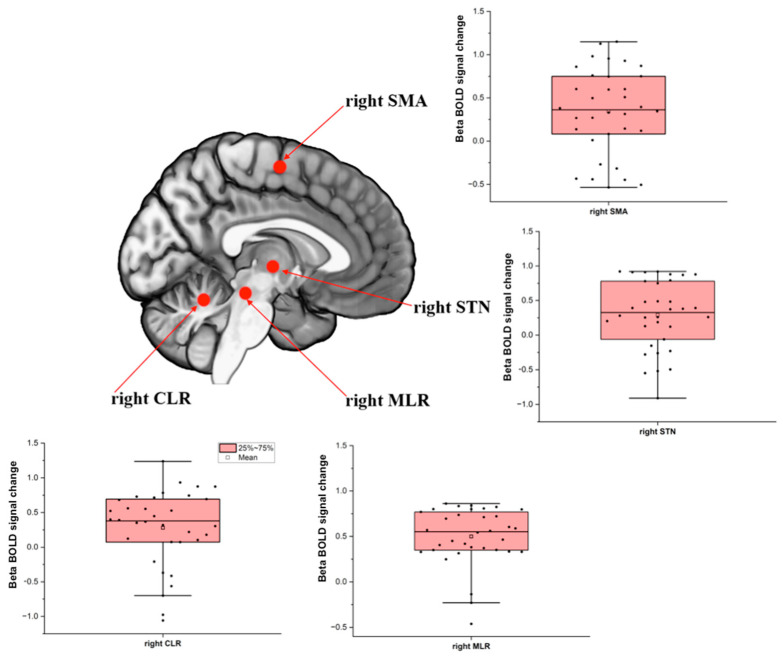
Box−plot showing the mean and distribution of the blood oxygenation level-dependent (BOLD) signal of the locomotor hubs during step initiation used as regions of interest for the multiple regression analysis to explain the loss of presynaptic inhibition during anticipatory postural adjustment. SMA = supplementary motor area; STN = subthalamic nucleus; MLR = mesencephalic locomotor region; CLR = cerebellar locomotor region.

**Figure 2 brainsci-14-00178-f002:**
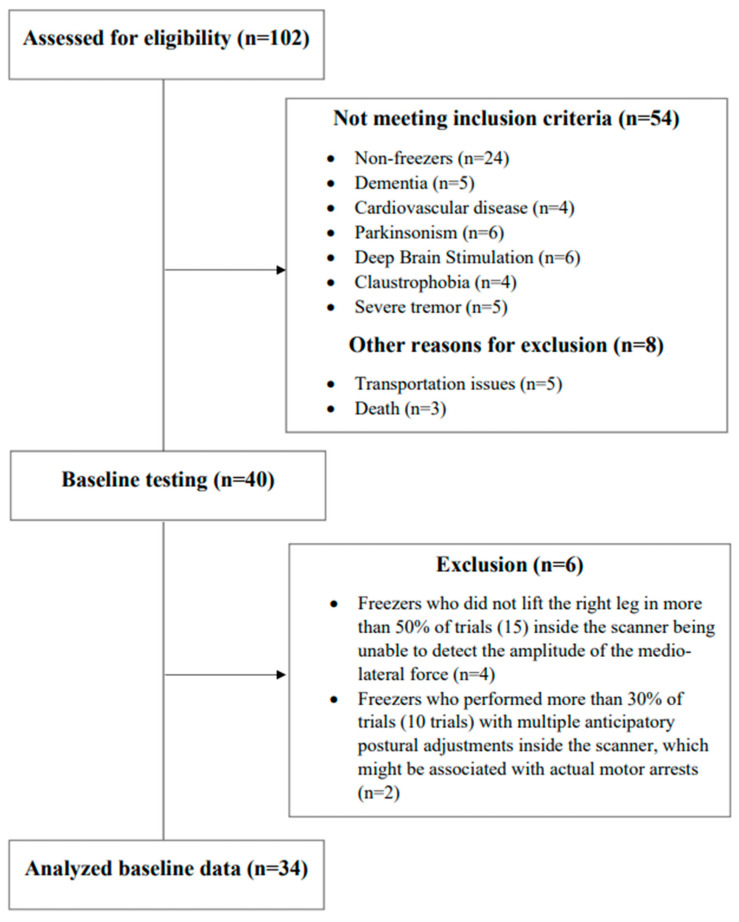
A schematic representation of participant recruitment and allocation.

**Figure 3 brainsci-14-00178-f003:**
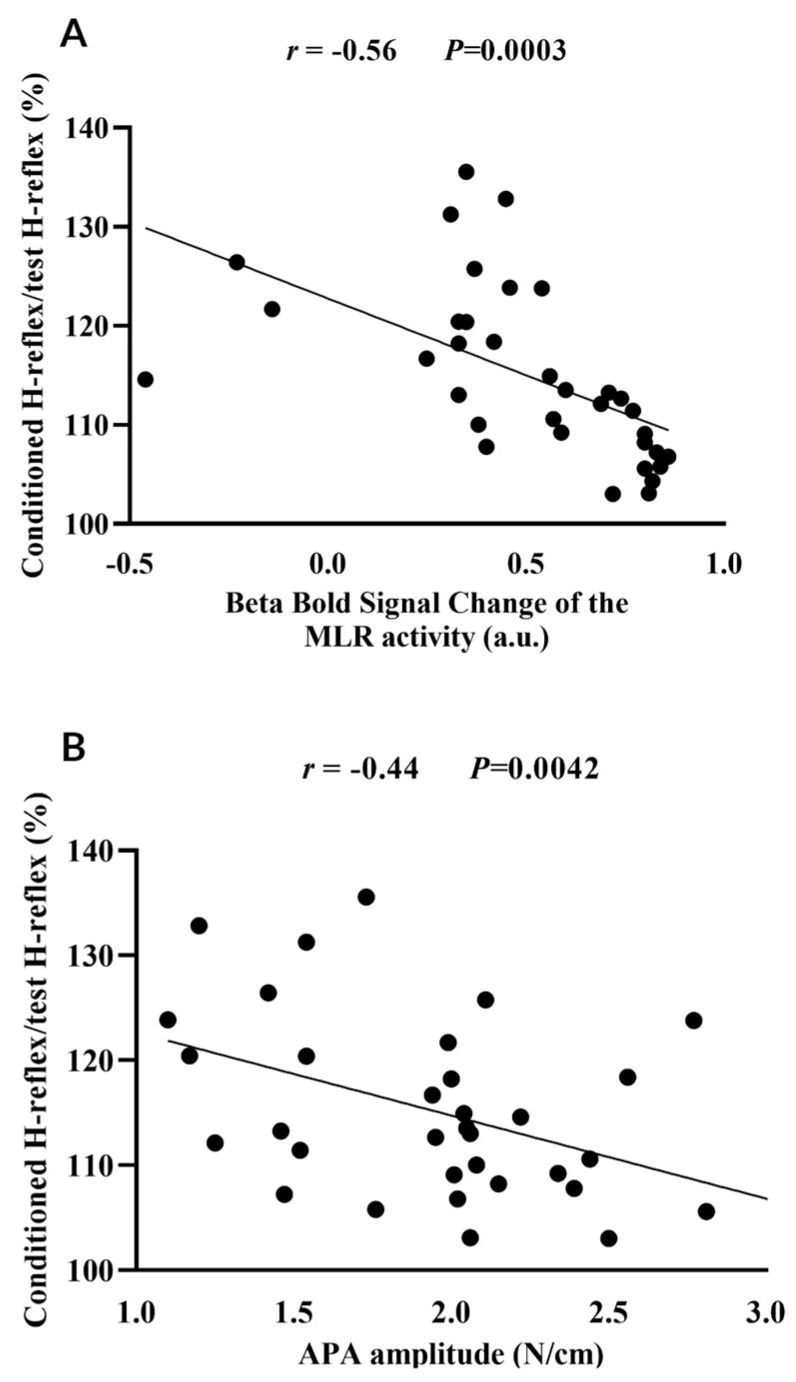
Correlation of loss of presynaptic inhibition of the soleus muscle (i.e., ratio of the conditioned H−reflex relative to the test H−reflex) with the beta of blood oxygenation level-dependent (BOLD) signal change of the mesencephalic locomotor region (MLR) (**A**) and with the anticipatory postural adjustment (APA) amplitude (**B**) during step initiation. a.u. = arbitrary units.

**Figure 4 brainsci-14-00178-f004:**
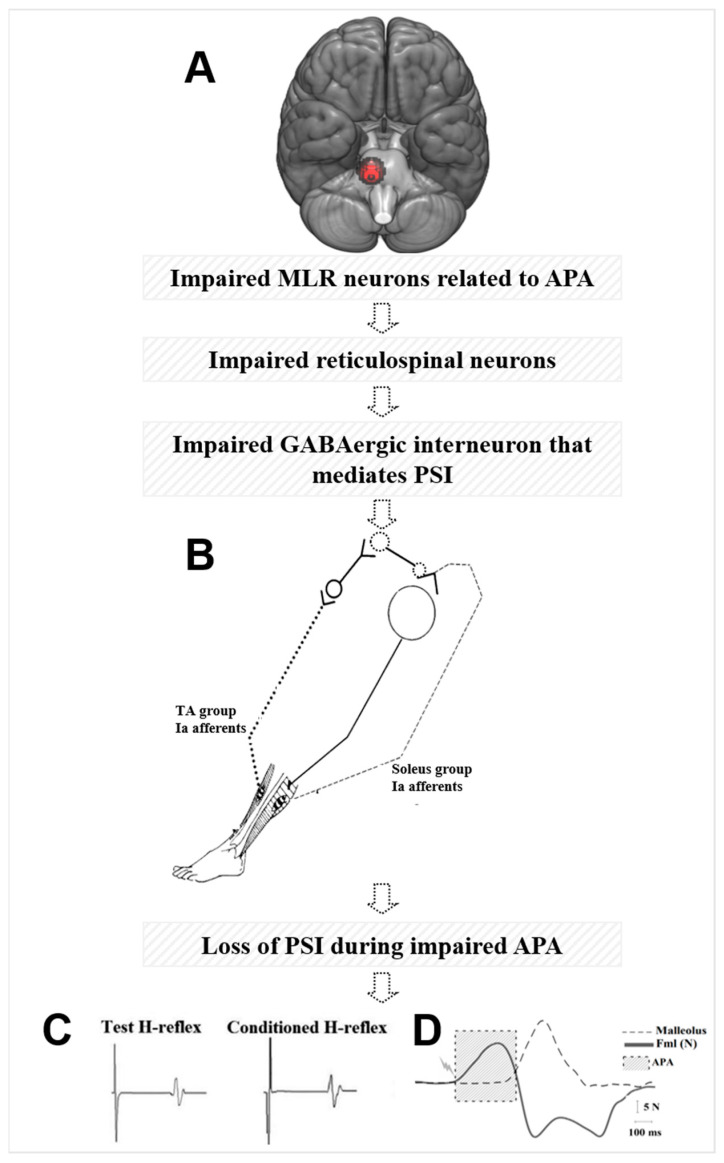
(**A**) Hypothetical model of the loss of presynaptic inhibition (PSI) during anticipatory postural adjustment (APA) via decreased beta of blood oxygenation level-dependent (BOLD) signal change of the right mesencephalic locomotor region (MLR) during step initiation in freezers. (**B**) PSI mechanism of Ia afferents induced by a conditioning stimulus on the common peroneal nerve. (**C**) Representative traces of test and conditioned H-reflexes (average over 15 responses) for a freezer during APA for step initiation. (**D**) Forward displacement of the reflective marker attached to the ankle during step is represented by dashed line, mediolateral force amplitude (Fml) during APA is represented by continuous line, and APA is represented by shaded square.

**Table 1 brainsci-14-00178-t001:** Characteristics of the freezers. Mean (SD).

	**Freezers (** **n** ** = 34)**	**Range**
**Demographics**		
Men/women (number)	26/8	-
Age (years)	63.7 (9.0)	49 to 80
Educational level (years)	10.1 (5.2)	4 to 24
Body mass (kg)	69.6 (11.2)	49 to 94
Height (cm)	1.5 (0.2)	1 to 2
Body mass index (kg/m^2^)	25.7 (3.1)	20 to 32
MMSE (score)	26.0 (1.6)	24 to 29
**Clinical variables**		
Years since diagnosis (years)	8.8 (5.1)	2 to 25
Hoehn and Yahr staging scale (a.u)	3.2 (0.4)	3 to 4
Number of participants in stage 3	28	-
Number of participants in stage 4	6	-
Symptom-dominant side (R/L/B)	0/6/28	
Right-legged (number)	20	
Left-legged (number)	14	
UPDRS-III (score)	49.9 (11.2)	23 to 67
PIGD (score)	8.6 (2.3)	4 to 13
NFOGQ (score)	22.2 (5.4)	12 to 28
Stroop Color-Word Test (a.u)	69.5 (42.9)	20.8 to 160.2
L-Dopa equivalent units (mg·day^−1^)	802.6 (270.7)	300 to 1300
**Behavioral variables**		
The ratio of the conditioned H-reflex relative to the test H-reflex (%)	115.1 (8.6)	103.0 to 135.6
FOG-ratio (a.u)	12.3 (10.8)	2.1 to 54.6
APA amplitude (N/cm)	1.9 (0.4)	1.1 to 3.0
APA duration (ms)	483.1 (59.9)	402.6 to 676.7
raEMG of the tibial anterior muscle (mV)	0.06 (0.03)	0.1 to 0.10
raEMG of the soleus muscle (mV)	0.08 (0.04)	0.02 to 0.19
Co-contraction ratio (%)	89.8 (58.9)	0 to 219.9
**Beta of the BOLD signal change**		
Beta of BOLD signal change of the right SMA (a.u.)	0.3 (0.4)	−0.5 to 1.1
Beta of BOLD signal change of the right STN (a.u.)	0.3 (0.5)	−1.0 to 0.9
Beta of BOLD signal change of the right MLR (a.u.)	0.4 (0.3)	−0.4 to 0.8
Beta of BOLD signal change of the right CLR (a.u.)	0.3 (0.5)	−1.1 to 1.2

Abbreviations: MMSE = Mini-Mental State Examination; R = right; L = left; B = both; UPDRS-III = Unified Parkinson Disease Rating Scale section III; PIGD = postural instability and gait disturbance; NFOGQ = New Freezing of Gait Questionnaire; FOG = freezing of gait; R = right; L = left; B = both; L-Dopa = Levodopa; a.u. = arbitrary units; APA = anticipatory postural adjustment; BOLD = blood oxygenation level dependent signal; SMA = supplementary motor area; STN = subthalamic nucleus; MLR = mesencephalic locomotor region; CLR = cerebellar locomotor region; and raEMG = rectified and averaged electromyography.

**Table 2 brainsci-14-00178-t002:** Factors (independent variables) and dependent variable (ratio of the conditioned H-reflex relative to the test H-reflex—presynaptic inhibition of the soleus muscle) were included in the linear multiple regression.

Independent Factors	Partial R^2^	Model R^2^ Change	*F* Value	*p* Value	Adjusted Model R^2^ Change *(Variance Explained)*
Beta of BOLD signal change of the MLR during step initiation (a.u)	0.3152	0.3152	14.73	0.0006	0.49
APA amplitude (N/cm)	0.1347	0.4500	7.59	0.0097	
FOG-ratio (a.u.)	0.0541	0.5041	3.28	0.0804	
Disease duration (years)	0.0549	0.5590	3.61	0.0674	

Abbreviations: BOLD = blood oxygenation level-dependent; MLR = mesencephalic locomotor; APA = anticipatory postural adjustment; FOG = freezing of gait; a.u. = arbitrary units.

## Data Availability

The data are not publicly available due to privacy restrictions. Data will be made available on request.

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
