# Peer review of "Mesencephalic Locomotor Region and Presynaptic Inhibition during Anticipatory Postural Adjustments in People with Parkinson’s Disease"

_brainsci, 2024, doi:10.3390/brainsci14020178_

Round 1

Reviewer 1 Report

Comments and Suggestions for Authors

In this prospective study of 34 PD patients with FOG, the investigators test the hypothesis that MLR activity as measured by BOLD activation during an APA task would explain their previously reported finding of loss of spinal inhibition represented by “PSI” in FOG. This is an interesting study, which adds a key link in the pathophysiology of FOG, linking brainstem pathophysiology with downstream spinal effects, and also linking functional brain imaging with physiological findings. Limitations are well reported. Conclusions are well drawn from the data presented. The manuscript is well written. 

Minor suggestions:

The concept of PSI may be foreign to the reader and may be misunderstood initially as referring to striatal presynaptic dopaminergic denervation in PD or presynaptic neurotransmitter release. A more fundamental description of the PSI early on, in the introduction may benefit the unfamiliar reader. 

Figure 4 needs a legend and further explanation of the bottom right figure. 

Reference 20 and 86 and duplicates. 

Author Response

Reviewer 1

In this prospective study of 34 PD patients with FOG, the investigators test the hypothesis that MLR activity as measured by BOLD activation during an APA task would explain their previously reported finding of loss of spinal inhibition represented by “PSI” in FOG. This is an interesting study, which adds a key link in the pathophysiology of FOG, linking brainstem pathophysiology with downstream spinal effects, and also linking functional brain imaging with physiological findings. Limitations are well reported. Conclusions are well drawn from the data presented. The manuscript is well written. 

Minor suggestions:

The concept of PSI may be foreign to the reader and may be misunderstood initially as referring to striatal presynaptic dopaminergic denervation in PD or presynaptic neurotransmitter release. A more fundamental description of the PSI early on, in the introduction may benefit the unfamiliar reader. 

Figure 4 needs a legend and further explanation of the bottom right figure. 

Reference 20 and 86 and duplicates. 

Answer: Thank you for your comments. We have included a concept of PSI earlier in the Introduction, as follow:

Presynaptic inhibition (PSI) is a spinal inhibitory mechanism often proposed to explain changes in the reflex pathways [1]. PSI mechanism involves GABAergic primary afferent depolarization interneurons in the spinal cord [1, 2]. PSI is responsible for modulating sensory feedback at the spinal level for walking [3] and postural preparation [4-8].

We have included a legend for Fig 4, as follow:

Figure 4. A) Hypothetical model of the loss of presynaptic inhibition (PSI) during anticipatory postural adjustment (APA) via decreased beta of blood oxygenation level-dependent (BOLD) signal change of the right mesencephalic locomotor region (MLR) during step initiation in freezers. B) PSI mechanism of Ia afferents induced by a conditioning stimulus on the common peroneal nerve. C) representative traces of test and conditioned H-reflexes (average over 15 responses) for a freezer during APA for step initiation. D) Forward displacement of the reflective marker attached to the ankle during step is represented by dashed line, mediolateral force amplitude (Fml) during APA is represented by continuous line, and APA is represented by shaded square.

Thanks for checking out the references, we have done a double check on it.

Reviewer 2 Report

Comments and Suggestions for Authors

The manuscript of Silva-Batista et al "Mesencephalic Locomotor Region and Presynaptic Inhibition during Anticipatory Postural Adjustments in People with Parkinson’s Disease" is interesting and well written. I have a few comments and questions.

1. It is not specified in the Methods that patients were studied ON med, it is mentioned as a limitation later. Suggest adding this to the methods, and I am not sure that examining patients ON med is a limitation, as it is ON period gait freezing (not OFF period bradykinesia) that is the phenomenon being studied, therefore ON med examination is appropriate.

2. The authors chose a an ISI of 100 ms for presynaptic inhibition of soleus H reflex by common peroneal nerve prepulse. Some earlier studies suggest that strongest PSI is achieved with slightly shorter ISI that 100 eg 60-90, why was 100 ms selected?

3. Table 1: UPDRS 3 data, is this OFF or ON med, please specify, and preferably include both OFF and ON med scores so the reader can have some idea about overall L-dopa responsiveness in this cohort.

4. It is noteworthy that the mean H conditioned/ H unconditioned was greater that 100% (around 115%) indicating  loss of PSI at group level in this corhort. This is evident from the Table but not stated explicitly in the Results text. Suggest adding a small paragraph with the results of H-reflex PSI studies to the text.

5. The discussion is very good overall and the pathophysiological explanations are a strength. However, I was puzzled by the statement (Page 10 lines 367-370) that loss of descending reticulospinal inhibition of GABAergic spinal interneurons would lead to loss of PSI. I would have expected disinhibition of inhibitory GABAergic spinal interneurons to increase spinal presynaptic inhibition. Can the authors explain this apparent discrepancy. 

Author Response

Reviewer 2

The manuscript of Silva-Batista et al "Mesencephalic Locomotor Region and Presynaptic Inhibition during Anticipatory Postural Adjustments in People with Parkinson’s Disease" is interesting and well written. I have a few comments and questions.

  1. It is not specified in the Methods that patients were studied ON med, it is mentioned as a limitation later. Suggest adding this to the methods, and I am not sure that examining patients ON med is a limitation, as it is ON period gait freezing (not OFF period bradykinesia) that is the phenomenon being studied, therefore ON med examination is appropriate.

Answer: Thank you for your comments. Details about ON medication has been included (lines 187-188). We have added your great viewpoint in the limitation section, as follows:

Although ON medication assessments do not reflect the true disease state and FOG is most commonly seen in the OFF-state [9], ON medication decreases bradykinetic effects on FOG episodes [10].

  1. The authors chose a an ISI of 100 ms for presynaptic inhibition of soleus H reflex by common peroneal nerve prepulse. Some earlier studies suggest that strongest PSI is achieved with slightly shorter ISI that 100 eg 60-90, why was 100 ms selected?

Answer: Thank you for this great point. It is a great discussion in the literature. Our previous works have shown that the H-reflex is strongly inhibited using a conditioning-test interval of 100 ms [7, 8]. Previous works have shown that conditioning-test intervals of <100 ms are likely to involve postsynaptic mechanisms, decreasing the ability to assess presynaptic influences [11]. Also, recommendations have been proposed for studies that use soleus H-reflex depression by common peroneal nerve stimulation at a motor threshold level, which is indicated for conditioning-test intervals of 60–120 ms [1, 12]. We have included this point in the manuscript.

  1. Table 1: UPDRS 3 data, is this OFF or ON med, please specify, and preferably include both OFF and ON med scores so the reader can have some idea about overall L-dopa responsiveness in this cohort.

Answer: Thank you for checking that. We have assessed patients only in the ON medication status for all measurements. We have clarified that in the manuscript (lines 187-188)

  1. It is noteworthy that the mean H conditioned/ H unconditioned was greater that 100% (around 115%) indicating loss of PSI at group level in this corhort. This is evident from the Table but not stated explicitly in the Results text. Suggest adding a small paragraph with the results of H-reflex PSI studies to the text.

Answer: The ratio of the conditioned H-reflex relative to the test H-reflex is included in Table 1 (range 103 to 135.6%), we also have included these values in the Results section, as requested (lines 508-509):

Freezers presented loss of PSI during APA for step initiation indicated by a high ratio of the conditioned H-reflex relative to the test H-reflex = 115.1(8.6).

  1. The discussion is very good overall and the pathophysiological explanations are a strength. However, I was puzzled by the statement (Page 10 lines 367-370) that loss of descending reticulospinal inhibition of GABAergic spinal interneurons would lead to lossof PSI. I would have expected disinhibition of inhibitory GABAergic spinal interneurons to increasespinal presynaptic inhibition. Can the authors explain this apparent discrepancy. 

Answer: Thank you for that. PSI is responsible for regulating the amplitude of H-reflex. Loss of PSI in freezers suggests that GABAergic interneurons are not activated to inhibit the conditioned H-reflex, which is facilitated (i.e., increased), then, freezers have a loss of PSI (high ratio of the conditioned H-reflex relative to the test H-reflex). We have included it in the manuscript, as follow:

As a result, the conditioned H-reflex is facilitated/increased (i.e., not inhibited), leading to loss of PSI during step initiation in freezers that are associated with impaired APA amplitude and FOG severity.

Reviewer 3 Report

Comments and Suggestions for Authors

The main question under consideration is the relationship between MLR activity, APA amplitude and PSI loss during step initiation in people with Parkinson's disease and gait faltering.

The topic is highly relevant and addresses a crucial issue in Parkinson's disease research - gait faltering. What it aims to uncover is the neurophysiological mechanisms contributing to this symptom, which shows a significant contribution to the field.

This research provides valuable insight into the interaction between MLR, APA and PSI, which may be unique compared to other published material. It presents a hypothesized model to explain the observed relationships.

The authors may consider providing additional information about the fMRI protocol and its validation for modeling pacing initiation. In addition, clarification on how a specific control was implemented could improve the methodology.

The conclusions are consistent with the data presented. The study establishes a link between MLR dysfunction, reduced APA amplitude, and loss of PSI contributing to gait faltering in Parkinson's disease.

The references seem to be appropriate.

In summary, the study is well conducted and provides insightful information. Improvements in the detail of the methodology could have improved the reliability of the study.

Author Response

Reviewer 3

The main question under consideration is the relationship between MLR activity, APA amplitude and PSI loss during step initiation in people with Parkinson's disease and gait faltering.

The topic is highly relevant and addresses a crucial issue in Parkinson's disease research - gait faltering. What it aims to uncover is the neurophysiological mechanisms contributing to this symptom, which shows a significant contribution to the field.

This research provides valuable insight into the interaction between MLR, APA and PSI, which may be unique compared to other published material. It presents a hypothesized model to explain the observed relationships.

The authors may consider providing additional information about the fMRI protocol and its validation for modeling pacing initiation. In addition, clarification on how a specific control was implemented could improve the methodology.

The conclusions are consistent with the data presented. The study establishes a link between MLR dysfunction, reduced APA amplitude, and loss of PSI contributing to gait faltering in Parkinson's disease.

The references seem to be appropriate.

In summary, the study is well conducted and provides insightful information. Improvements in the detail of the methodology could have improved the reliability of the study.

Answer: Thank you for your comments. Since the fMRI protocol is too long and we have published previously the validation and methodological details (Brain networks associated with anticipatory postural adjustments in Parkinson's disease patients with freezing of gait Andrea C de Lima-Pardini 1Daniel B Coelho 2Mariana P Nucci 3Catarina C Boffino 4Alana X Batista 3Raymundo M de Azevedo Neto 5Carla Silva-Batista 6Egberto R Barbosa 7Rajal G Cohen 8Fay B Horak 9Luis A Teixeira 10Edson Amaro Jr 3PMID: 33395957PMCID: PMC7575874 DOI: 10.1016/j.nicl.2020.102461; An fMRI-compatible force measurement system for the evaluation of the neural correlates of step initiation Andrea Cristina de Lima-Pardini 1Raymundo Machado de Azevedo Neto 1Daniel Boari Coelho 2Catarina Costa Boffino 1Sukhwinder S Shergill 3Carolina de Oliveira Souza 4Rachael Brant 4Egberto Reis Barbosa 4Ellison Fernando Cardoso 1Luis Augusto Teixeira 2Rajal G Cohen 5Fay Bahling Horak 6Edson Amaro Jr 1 PMID: 28230070 PMCID: PMC5322382 DOI: 10.1038/srep43088), we have included in the Supplementary file a detailed summary of these descriptions.

Reviewer 4 Report

Comments and Suggestions for Authors

The major idea of the study, the Methods section, and the result are clear. Most of my comments were anticipated in the “Future directions” and “Limitations” sections. Still, I have several minor comments to this manuscript:

1.      Wouldn’t it be helpful to view a BOLD/fMRI image of the studied brain regions (MLR, CLR, STN, and SMA) during APA (along with the Figure 1)?

2.      Item 63 in the reference list (line 677) – name of one author and the journal title are missing. Looks like the following: Shik ML, Severin FV, OrlovskiÄ­ GN. Upravlenie khod'boÄ­ i begom posredstvom elektricheskoÄ­ stimulatsii srednego mozga [Control of walking and running by means of electric stimulation of the midbrain]. Biofizika. 1966;11(4):659-66. Russian (from PubMed).

3.      Line 199 – a study is referenced as a name, not as number.

4.      PD indeed starts asymmetrically (as far as I know, usually with tremor in the right hand). Correspondingly, PD should be manifested mostly in the left side brain circuitries, not in the right side (line 215)? In that respect, were at least some of the subjects left-handed (left-legged) or all of them were right-handed? Some normal subjects initiate walking (jumping) with the left leg.

5.      Line 140 – please clarify, which leg (the left or the right) actually initiated stepping? Probably, the right leg (line 144). On the other hand, the left leg was tested (line 213)?

6.      The problem of stepping inhibition in PD (freezing) to some extend can be explained by behavioral issues. PD patients are often over-cautious and stay motionless because they simply afraid to fall. (e.g.  M.Latash, Neurophysiological basement of movements, 1996, R. J. Peterka, "Sensorimotor integration in human postural control", Journal of Neurophysiology, vol. 88, pp. 1097-1118, DOI:10.1152/jn.2002.88.3.1097 2002. P. F. Smith, "Vestibular functions and Parkinson's disease", Frontiers in Neurology, vol. 9. â„–1085. doi: 10.3389/fneur.2018.01085., 2018). This aspect can be further discussed (in addition to behavioral variables presented in this study).

Author Response

Reviewer 4

The major idea of the study, the Methods section, and the result are clear. Most of my comments were anticipated in the “Future directions” and “Limitations” sections. Still, I have several minor comments to this manuscript:

  1. Wouldn’t it be helpful to view a BOLD/fMRI image of the studied brain regions (MLR, CLR, STN, and SMA) during APA (along with the Figure 1)?

Answer: Unfortunately, we have only the extracted beta BOLD signal change for each ROI since our study was finalized in April 2019. We cannot run the data to generate BOLD/fMRI image since the fMRI data was removed from the workstation. Keeping the data processed for almost 5 years has a high cost on the workstation and we had funding for storage of the acquired and processing data only for 3 years. We have included box-plots along with Figure 1 for visualization of data distribution.

  1. Item 63 in the reference list (line 677) – name of one author and the journal title are missing.Looks like the following: Shik ML, Severin FV, OrlovskiÄ­ GN. Upravlenie khod'boÄ­ i begom posredstvom elektricheskoÄ­ stimulatsii srednego mozga [Control of walking and running by means of electric stimulation of the midbrain]. Biofizika. 1966;11(4):659-66. Russian (from PubMed).

Answer: Thank you for checking that. We have changed it.

  1. Line 199 – a study is referenced as a name, not as number.

Answer: Thank you for checking that. We have changed it

  1. PD indeed starts asymmetrically (as far as I know, usually with tremor in the right hand). Correspondingly, PD should be manifested mostly in the left side brain circuitries, not in the right side (line 215)? In that respect, were at least some of the subjects left-handed (left-legged) or all of them were right-handed? Some normal subjects initiate walking (jumping) with the left leg.

Answer: Thank you for that. We have included the dominant side in Table 1 as well as the symptom-dominant side. Participants were instructed to initiate the step with the right leg in all conditions. Thus, the APA task (fMRI protocol and step initiation on force platform) was lateralized to the left (support) leg only. We assessed the APA in the left leg because the participants had either both sides affected (moderate to severe PD - stages 3 and 4) or the left side affected. Previous studies from our and other groups have demonstrated that freezers tend to show predominant involvement of right-sided brain circuitry [13-16] which reinforces the importance of the APA task lateralized to the left leg.

  1. Line 140 – please clarify, which leg (the left or the right) actually initiated stepping? Probably, the right leg (line 144). On the other hand, the left leg was tested (line 213)?

Answer: Yes, the participants initiated stepping with the right leg, thus, only the left leg was assessed in both fMRI and force platform. We have improved this information in the manuscript.

  1. The problem of stepping inhibition in PD (freezing) to some extend can be explained by behavioral issues. PD patients are often over-cautious and stay motionless because they simply afraid to fall. (e.g.  M.Latash, Neurophysiological basement of movements, 1996, R. J. Peterka, "Sensorimotor integration in human postural control", Journal of Neurophysiology, vol. 88, pp. 1097-1118, DOI:10.1152/jn.2002.88.3.1097 2002. P. F. Smith, "Vestibular functions and Parkinson's disease", Frontiers in Neurology, vol. 9. â„–1085. doi: 10.3389/fneur.2018.01085., 2018). This aspect can be further discussed (in addition to behavioral variables presented in this study).

Answer: We have included this discussion as follows:

It is important to highlight that older people and individuals with PD are often over-cautious and stay motionless due to fear of falling [17, 18], which could impact PSI levels during step initiation. Increased postural stiffness and agonist–antagonist contraction is required to maintain postural stability during afraid situations [19]. PSI is known to increase during agonist–antagonist contraction and muscle stiffness during an upright stance in challenging situations (e.g., eyes closed) [20]. Since freezers have loss of PSI during step initiation and we did not find any association between co-contraction and loss of PSI, we believe that behavioral issues did not influence the PSI levels during step initiation in freezers. Thus, loss of PSI during step initiation may be due to abnormal MLR activity.